# Factors Affecting Health-Related Quality of Life among Healthcare Workers during COVID-19: A Cross-Sectional Study

**DOI:** 10.3390/medicina59010038

**Published:** 2022-12-24

**Authors:** Gyehyun Jung, Jihyun Oh

**Affiliations:** 1Department of Nursing, Jeonbuk Science College, Jeongeup 56204, Republic of Korea; 2Department of Nursing, College of Nursing and Health, Kongju National University, Kongju 32588, Republic of Korea

**Keywords:** health-related quality of life, sleep quality, depression, stress, anxiety

## Abstract

*Background and Objectives:* Healthcare workers are threatened by psychological well-being and mental health problems in disasters related to new infectious diseases, such as COVID-19, and this can also have a negative impact on health-related quality of life. Health-related quality of life of healthcare workers should not be neglected because it is closely related to patient safety. This study aimed to identify the relationship between mental health problems, psychological safety, sleep quality, and health-related quality of life of healthcare workers and factors that influence health-related quality of life during the COVID-19 pandemic. *Materials and Methods:* Data were collected from 301 healthcare workers working in five general hospitals with more than 300 beds in two provinces from 5 July 2021 to 16 July 2021. Data were analyzed using SPSS WIN 27.0. The data were analyzed using *t*-test, one-way analysis of variance, and stepwise multiple regression. *Results:* Our results showed that there was a significant difference in regular exercise, religion, economic status, and sleep quality. The DASS-21 stress, economic status, and alcohol consumption were factors affecting the total health-related quality of life. In the subcategories, the physical component score was influenced by DASS-21 stress and economic status, while the mental component score was influenced by DASS-21 depression, economic status, alcohol consumption, and sleep quality. *Conclusions:* Health care workers need continuous and active monitoring of their health level and quality of life, as they are at a risk of increasing work burden and infection due to prolonged exposure to COVID-19 as well as mental health issues such as stress and depression. Additionally, at the individual level, active participation in various programs that can raise awareness of health-related quality of life along with physical health promotion activities should be encouraged. At the organizational level, it is necessary to prepare a compensation system, such as adjusting the workload of healthcare workers and ensuring break time; at the government level, disaster-related policies are needed to ensure a safe working environment for health care workers.

## 1. Introduction

The rapid spread of coronavirus (SARS-CoV-2; COVID-19) has resulted in unprecedented workload and growing responsibilities for the healthcare sector and healthcare workers worldwide [1,2]. Healthcare workers play an important role in responding to infectious diseases, preventing their spread, and directly engaging in treatment for emerging infectious diseases [2,3,4].

Early in the COVID-19 pandemic, healthcare workers were stressed by additional pandemic-related factors such as a high infection risk, a sudden surge in the number of cases, scarce personal protection equipment, many deaths, self-isolation, strenuous and challenging working hours, ambiguity about the protection provided by the vaccine, and stigmatization [5,6,7].

Healthcare workers have different health problems from the general public, and mortality rate from factors other than diseases is high [8]. Moreover, as COVID-19 continues to evolve, treatment protocols have been established, and attempts are being made to address poor psychological well-being such as mental health problems such as work-related stress and anxiety associated with infectious diseases among healthcare workers [5,9,10,11]. However, emerging infectious diseases, such as COVID-19, further threaten the mental health of healthcare workers and instigate conditions such as depression and post-traumatic stress disorder (PTSD).

In addition, the consequences of the present pandemic affect not only healthcare workers’ physical health but also their psychological and mental well-being [1]. Continuous exposure to work-related stress and mental health issues can impair well-being and the ability to work, contributing to poor patient safety, quality of care, and early retirement [5,7]. Furthermore, these problems can adversely affect health workers’ health-related quality of life, including disconnected relationships, problematic use of alcohol and other drugs, and suicidal thoughts [2,12,13].

Psychological safety represents a worker’s perception of how coworkers can respond to behaviors implied by interpersonal situations [14]. Psychological safety can be an important mechanism for reducing stress by creating an atmosphere of trust and risk-free communication [15]. During the COVID-19 pandemic, healthcare workers may have reduced health-related quality of life due to emotional exhaustion, but emotional exhaustion can be improved through psychological safety [16,17]. It can be assumed that emotional safety can improve the health-related quality of life of healthcare workers, and it is necessary to confirm this.

Many healthcare workers experience symptoms of sleep problems, including low sleep quality and short sleep duration, when faced with various threats from emerging infectious diseases [18]. Sleep problems, such as stress and depression, can reduce mental health and work efficiency [19,20]. Insufficient and poor sleep quality can be important early signs of underlying physical or mental health problems for healthcare workers as well as affect their health-related quality of life [20,21,22].

Therefore, it is necessary to investigate the mental health problems and health-related quality of life of healthcare workers in detail. It is also necessary to establish a theoretical basis for the basic data necessary for the development of intervention programs to improve health-related quality of life for healthcare workers in disaster situations related to emerging infectious diseases such as COVID-19. 

This study aims to identify the relationship between mental health problems, psychological safety, sleep quality, and health-related quality of life of healthcare workers and the factors that influence health-related quality of life during the COVID-19 pandemic.

## 2. Materials and Methods

### 2.1. Research Design and Subjects

This cross-sectional study aimed to identify the health quality of life of healthcare workers and the factors that influence it during the COVID-19 pandemic. A convenience sample of the respondents of this study were healthcare workers working in five general hospitals with more than 300 beds in Chungcheng and Jeolla provinces. The inclusion criteria were as follows: (1) Health and medical services personnel who have obtained qualifications and licenses and are permitted to engage in health and medical services as stipulated by the Health Care Laws and Regulations in the Framework Act on Health and Medical Services, and healthcare medical services personnel include healthcare professionals such as medical personnel, medical technicians, pharmacists, and those who are over 20 years old and currently working in hospitals during the COVID-19 pandemic; (2) those who do not have communication impairments and are conscious, understand the purpose of this study, and want to participate in the study; and (3) those who can read and understand the Korean language for the purpose of the survey. The exclusion criteria were as follows: health and medical service personnel who worked in public health centers or industries other than hospitals during the COVID-19 pandemic.

The minimum number of samples was 230 when set to a significance level (a) of 0.05, statistical power (1-β) of 95%, effect size (f) of 0.15, and 22 predictable variables [23]. Recruitment accounted for a 20% dropout rate among the study participants. The number of participants in this study was 301, which was greater than 230, suggesting that the statistical power was good. 

### 2.2. General Characteristics

The participants’ general characteristics were examined by age, sex, education level, marital status, regular exercise, smoking, alcohol consumption, religion, living arrangement, and economic status. Regular exercise means that adults should exercise for at least 150 min per week.

### 2.3. Health-Related Quality of Life

The Korean version of the Short Form—12 (SF-12) is an HRQOL measurement tool that was developed by Ware et al. [24]. SF-12 is an abbreviation of the Short Form—36 Health Survey Questionnaire (SF-36) and consists of twelve questions in eight areas for measuring quality of life (physical functioning, limitation in physical role, pain, general health, vitality, social function, limitation in emotional role, and mental health). Each physical component score (PCS) and mental component score (MCS) was 50 points, and the sum of the two components ranged from 0 to 100 points; the higher the score, the better the health and quality of life. The reliability of the SF-12 at the time of development was Cronbach’s alpha = 0.89. In this study, Cronbach’s alpha was 0.91.

### 2.4. Psychological Safety

Psychological safety is the degree to which members feel safe when expressing ideas or opinions in an organization [25]. Psychological safety was measured using seven items from the team’s psychological safety scale developed by Edmondson [25]. The main questions included, “Even if I make a mistake, my organization will not reject me”, and “In my organization, anyone can raise difficult problems or uncomfortable issues openly”. It consists of a 5-point Likert scale, with 1 point indicating “strongly disagree” and 5 points indicating “strongly agree”. The score ranges from 5 to 35 points, where the higher the score, the higher the team’s psychological safety. The Cronbach’s alpha at the time of development was 0.82. The Cronbach’s alpha in this study was 0.92.

### 2.5. Sleep Quality

The Pittsburgh Sleep Quality Index (PSQI) as developed by Buysse et al. was used to measure sleep quality [26]. The PSQI consists of nineteen items and seven subdomains: subjective sleep quality, sleep efficiency, sleep disturbance, sleep latency, sleep duration, use of sleeping medication, and daytime dysfunction. Each item was scored from 0 to 3 points, and the total scores of the seven subdomains ranged from 0 to 21. A higher total PSQI score indicates poor sleep quality. Based on the criteria suggested by Buysse et al. [26], with a score of 5 as the standard, the groups were divided into a good sleep-quality group (the total scores of the PSQI were less than five) and a bad sleep-quality group (when the score was five or higher). The Cronbach’s alpha in this study was 0.90.

### 2.6. The Depression, Anxiety, and Stress Scale–21 (DASS–21)

Depression, anxiety, and stress were measured using the Korean version of the Depression, Anxiety, and Stress Scale (K-DASS-21). The DASS-21 has twenty-one items, with seven in each of the three subscales (depression, anxiety, and stress). The items asked about depressive symptoms (e.g., feeling downhearted and blue), anxiety symptoms (e.g., feeling panicky), and general stress symptoms (e.g., the tendency to overreact to situations). Response options were recorded on a 4-point scale (from 0 = did not apply to me at all, to 3 = applied to me most of the time). Each subscale consists of seven items, and the total score ranges from 0 to 21. Higher scores indicate greater psychological distress [27]. The scale’s reliability and validity were verified with Cronbach α = 0.87 for depression, 0.83 for anxiety, and 0.83 for general stress in a study by Lee et al. (2011) [28]. Internal consistency coefficients for depression, 0.82 for anxiety, and 0.86 for general stress were 0.85, 0.82, and 0.85, respectively.

### 2.7. Data Collection and Period

Data on 301 healthcare workers were collected from 5 July 2021 to 16 July 2021. The survey was completed in approximately 20 min and was conducted once. Before proceeding with the study, we explained its purpose to the head of the nursing department and the persons in charge of the departments of the hospitals and obtained permission to collect data. The researchers explained the purpose and method of the study, confidentiality, and the time required to complete the survey in person to the study subjects and then requested their written informed consent to voluntarily participate in the study by completing the survey. It was explained that the surveys collected from the participants were anonymous and would not be used for any purpose other than this study. Furthermore, the subjects could withdraw from this study at any time upon reconsideration that they did not wish to participate in the study, and this would not cause any disadvantages.

### 2.8. Ethical Considerations

Consent was obtained from all study participants before collecting data from university students in South Korea. Data collection was conducted after the Institutional Review Board approved the tools and survey process used by the participants at Daejeon University (1040647-202101-HR-004-02). This study obtained permission from the directors of nursing and the heads of the nursing departments of five general hospitals. Subsequently, the researchers explained the necessity and method of the study to the healthcare workers, and consent to participate in the study was obtained. Participants who provided written informed consent to participate in this study completed a self-reported survey that took approximately 20 min. The tools used in the surveys were widely and universally used; therefore, the surveys were conducted safely. Additionally, the collected surveys were anonymous and could not be used for any purpose other than this study. If the subjects did not want to participate in the study, they could withdraw at any time and were informed that negative consequences would not occur.

### 2.9. Data Analysis

Data were analyzed using SPSS WIN 27.0; IBM Corp., Armonk, NY, USA), and the specific analysis was as follows: The common characteristics of the study subjects and the difference in quality of life according to their general characteristics were analyzed using the *t*-test, ANOVA, and post hoc analysis using the Scheffé test. The independent variables of this study, DASS-21, psychological safety, sleep quality, and health-related quality of life, were analyzed using mean and standard deviation. Correlations between the DASS-21, psychological safety, sleep quality, and health-related quality of life of the study participants were analyzed using the Pearson’s correlation coefficient. Stepwise multiple regression was used to identify variables affecting the quality of life.

## 3. Results

### 3.1. Participants’ General Characteristics

The study distributed surveys to 319 participants, and 301 surveys were included in the analysis (response rate of 94.3%), excluding 18 surveys whose responses were insincere or lacking (Table 1). The average age of the study subjects was 37.79 (9.75). Women comprised 86.7%, nurses comprised 74.8%, married women comprised 57.8% (more than half), and 75.1% were college graduates. Most participants responded that they did not exercise regularly (70.8%), 90.7% did not smoke, 53.2% did not drink, and 51.8% had a religion. Those who lived with family members were 79.7%; 83.4% answered that their economic status was “medium” or higher; 31.6% had good sleep quality with a score less than 5 points; and 68.4% had poor sleep quality with a score over 5 points.

### 3.2. Differences in Health-Related Quality of Life

Differences in total health-related quality of life according to the subjects’ general characteristics were analyzed. The total health-related quality of life differed according to regular exercise (*t* = 2.370, *p* = 0.018), religion (*t* = 1.982, *p* = 0.048), economic status (*t* = 2.261, *p* = 0.024), and sleep quality (*t* = 2.612, *p* = 0.009). Differences in general characteristics according to the two health-related quality-of-life subdomains were also analyzed. PCS scores according to general characteristics differed depending on regular exercise (*t* = 2.452, *p* = 0.015), economic status (*t* = −2.006, *p* = 0.046), and sleep quality (*t* = 2.178, *p* = 0.030). The MCS, according to general characteristics, differed depending on alcohol consumption (*t* = 2.299, *p* = 0.046), religion (*t* = −2.171, *p* = 0.031), and sleep quality (*t* = 2.312, *p* = 0.021). 

### 3.3. DASS-21, Psychological Safety, Sleep Quality, and Health-Related Quality of Life

The average values of each variable are listed (Table 2). The mean total health-related quality of life was 66.62 (7.39), and of the two subdomains of the total health-related quality of life, PCS was 61.22 (6.86), and MCS was 72.01 (10.18). DASS-21 consisted of three subdomains: the mean of DASS-21 depression was 4.16 (3.72), the mean of DASS-21 anxiety was 2.83 (3.22), and the mean of DASS-21 stress was 5.92 (3.72). The average psychological safety was 22.34 (3.48). The average total sleep quality was 6.88 (3.56), which was higher than 5 points, indicating poor sleep quality.

### 3.4. Correlations between DASS-21, Psychological Safety, Sleep Quality, and Health-Related Quality of Life

Table 3 shows the correlations between DASS-21, psychological safety, sleep quality, and health-related quality of life. Health-related quality of life was negatively correlated with DASS-21 depression (r = −0.325, *p* < 0.001), DASS-21 anxiety (r = −0.294, *p* < 0.001), DASS-21 stress (r = −0.354, *p* = 0.015), and sleep quality (r = −0.227, *p* <0.001). Higher health-related quality of life was significantly and weakly linked to lower levels of all the following: DASS-21 depression, anxiety, stress, and sleep quality. DASS-21 depression positively correlated with DASS-21 anxiety (r = 0.781, *p* < 0.001), DASS-21 stress (r = 0.841, *p* < 0.001), and sleep quality (r = 0.388, *p* < 0.001). That is, higher levels of depression were significantly associated with positive correlations with higher rates of anxiety and stress and lower sleep quality. A higher DASS-21 depression score was negatively correlated with psychological safety (r = −0.259, *p* < 0.001). Specifically, the higher the depression score, the lower the psychological safety score. Higher DASS-21 anxiety was associated with higher DASS-21 stress (r = 0.762, *p* < 0.001) and lower sleep quality (r = 0.366, *p* < 0.001) and was associated with lower psychological safety (r = −0.236, *p* < 0.001). Higher DASS-21 stress showed statistically significant negative correlations with less psychological safety (r = −0.207, *p* < 0.001), and higher stress was associated with lower sleep quality (r = 0.483, *p* < 0.001).

### 3.5. Factors Influencing the Health-Related Quality of Life of the Subjects

To identify the factors influencing health-related quality of life, variables that indicated differences in health-related quality of life among the demographic characteristics of the participants and variables that showed significant correlations with health-related quality of life were added to the regression equation, and stepwise multiple regression was evaluated (Table 4). The factors influencing the total quality of life of the subjects with statistical significance were DASS-21 stress (B = −0.792, SE = 0.106), economic status (B = 3.929, SE = 1.058), and drinking (B = −1.990, SE = 0.779), and the adjusted R squared (R^2^) was 0.173, indicating that the explanatory power of this model was 17.3%. The factors influencing the PCS of the subjects were DASS-21 stress (B = −0.474, SE = 0.104) and economic status (B = 2.954, SE = 1.041), with statistical significance, and the adjusted R squared (R^2^) was 0.071, indicating that the explanatory power of this model was 7.1%. Analysis of the factors influencing the MCS of the participants revealed that DASS-21 depression (B = −0.950, SE = 0.158), economic status (B = 6.045, SE = 1.468), drinking (B = −3.162, SE = 1.069), and sleep quality (B = −0.472, SE = 0.163) were statistically significant factors, with an adjusted R squared (R^2^) of 0.192, indicating that the explanatory power of this model was 19.2%.

## 4. Discussion

Healthcare workers’ health-related quality of life is very important not only for individual health but also for the maintenance of the medical system and the safety of patients. However, as COVID-19 continues, health-risk environments, such as the risk of infection exposure and exhaustion of healthcare workers, will also continue. Therefore, this study investigated the factors influencing healthcare workers’ health-related quality of life. In this study, health-related quality of life differed according to the general characteristics of healthcare workers, including regular exercise, religion, economic status, and sleep quality. The factors affecting healthcare workers’ health-related quality of life were stress, economic status, and alcohol consumption. In subcategories, physical health-related quality of life was influenced by stress and economic status, and mental health-related quality of life was influenced by depression, economic status, alcohol consumption, and sleep quality.

Health-related quality of life differed according to the general characteristics of healthcare workers, regular exercise, and religion. This is similar to the results of previous studies [29,30]. Religion is closely related to psychological aspects and mental health [29], and regular exercise is effective for emotional stability as well as physical health [30]; therefore, it would have been positively correlated with the health-related quality of life of healthcare workers. In particular, regular exercise may help improve fitness, prevent medical conditions, and reduce stress [31]. Therefore, it is necessary to consider activating physical activity and fitness programs for employees.

In this study, economic status affected health care workers’ health-related quality of life. This is similar to the results of previous studies [30,32]. This would have included psychologically distress regarding changes in financial conditions, such as salary cuts and job insecurity due to the economic recession caused by the COVID-19 pandemic [30], which would have affected the health-related quality of life of healthcare workers. Medical institution managers and drafters should strive to create a disaster management system that ensures that the working environment and employment opportunities of healthcare workers remain stable even in disaster situations [33].

Moreover, alcohol consumption affects healthcare workers’ mental health-related quality of life. This is similar to the results of previous studies [4,9,34]. According to Hennein and Lowe’s study [35], the possibility of alcohol-use disorder in healthcare workers was 42.6%, and they reported that the subjects did not control alcohol consumption well in the context of infectious diseases. In a study by Beiter et al. [9], healthcare workers with a high risk of increasing alcohol consumption reported great avoidance and psychological distress. This would have affected mental health-related quality of life, as healthcare workers could reduce tension and alleviate psychological distress through alcohol [4]. It is urgent to mediate the psychological outcomes of healthcare workers related to emerging infectious diseases, and the roles of the government and leaders are important to provide effective treatment in difficult clinical environments. 

On the other hand, stress and depression were the main factors affecting the health-related quality of life in this study. This finding is similar to the results of previous studies [12,13,36,37]. The degrees of anxiety, stress, and depression were lower than those reported in previous studies [6,9]. As mentioned in the study by Park [38], it can be said that he followed the principle of “information disclosure” learned through the MERS-Cov crisis in 2015. Information disclosure, fact checks, and quarantine promotion of the Korea Centers for Disease Control and Prevention, which are announced every day, would have been a great help to the success of quarantine as well as securing the trust of healthcare workers. According to previous studies [10,13,34], the younger and female, the less supported by their family, or the less supported by their workplace that people are, the higher the depression, the more the workload increases over time, and the more they are stressed by physical exhaustion and frequent death. These work-related psychological stresses and mental health symptoms would have a negative impact on the health-related quality of life of healthcare workers [12]. Therefore, to improve the psychological stress of healthcare workers and the health-related quality of life related to the emerging infectious disease, medical institutions and managers should actively consider intervention programs such as telehealth support, behavioral group therapy, cognitive behavioral therapy (CBT), and mindfulness-based therapy mentioned in previous studies to apply to various situations [36]. In addition, the disaster response system should include intervention plans for the psychological problems of healthcare workers.

In this study, sleep quality negatively affected the mental health-related quality of life. This is similar to the results of previous studies [2,7]. Sleep problems negatively affect the immune response by interfering with the daily rhythm of the body, increasing the sensitivity to infection, and having a strong influence on mental health problems such as stress and depression [19,20]. These sleep problems may have reduced the mental health-related quality of life of healthcare workers. On the other hand, reduced sleep quality can impair cognitive function and weaken decision-making ability, reduce clinical work efficiency, and increase the risk of medical errors [21]. Therefore, providing screening and monitoring programs to detect fundamental health conditions and treatments will improve the health of medical workers and improve their health directly or indirectly [22].

### 4.1. Implication

Based on the results of this study, the following implications are suggested to enable healthcare professionals to improve the quality of life of emerging infectious diseases such as COVID-19. First, it means that mental health and health-related quality of life of healthcare workers are closely related, and through this, it is necessary to identify the mental health-related factors of a job environment, such as stress and depression, and apply an intervention plan to alleviate them. Second, the guidelines for protection, facility expansion, and infection prevention education should be improved by health managers to prepare for new pandemics such as COVID-19 and prevent infection. Third, to control stress in healthcare workers, it is necessary to participate in appropriate sleep and physical activities at the individual level. At the organizational level, compensation systems such as workload control, management, and rest time guarantees should be prepared. Fourth, health care workers are often indifferent to their mental health [9]. Therefore, publicity and education are needed to increase awareness and understanding of mental health and health-related QoL. Fifth, considering that it is difficult for healthcare workers to gather in groups because of the nature of infectious disease disasters and difficulty in participating in face-to-face services, it is necessary to provide various non-face-to-face services, such as platforms that reflect the specificity of healthcare workers. Finally, the government should develop policies to stabilize employment and improve wages and welfare [30]. Additionally, it is necessary to develop and organize disaster management plans to protect the working environment in the event of future national disasters.

### 4.2. Limitation

This study had several limitations. First, the number of respondents in this study exceeded the minimum number of required samples; however, this was not conducted for all health managers in Korea. Second, because health-related quality of life in healthcare workers was assessed using a self-reported questionnaire, it may not be adequate to explore actual quality-of-life patterns. Therefore, the generalization of the results of this study to all healthcare workers should be considered carefully.

## 5. Conclusions

This study investigated the factors affecting the health-related quality of life of healthcare workers. Physical health-related quality of life was affected by stress and economic status, and mental health-related quality of life was affected by depression, economic status, alcohol consumption, and sleep quality. 

Based on the above results, it is necessary to maintain appropriate health and quality of life in order to perform related tasks, such as medical activities, as healthcare workers are in close contact with patients while performing their duties. Healthcare workers are at risk of increasing work burden and infection due to prolonged COVID-19 as well as mental health issues such as stress and depression. Therefore, continuous and active monitoring is needed to improve the health and quality of life of healthcare workers, and it is urgent to improve the healthcare system to prepare effective and efficient management measures at the individual and organizational levels. Moreover, when establishing a policy, the opinions of the field workers should be considered and reviewed.

## Figures and Tables

**Table 1 medicina-59-00038-t001:** Descriptive statistics and health-related quality of life according to general characteristics among participants (*N* = 301).

Variable	*N* (%)	Mean (SD)	Total HRQOL Score		PCS		MCS	
			Mean	*t* or F (*p*) Scheffé	Mean	*t* or F (*p*) Scheffé	Mean	*t* or F (*p*) Scheffé
Mean age (years) (range)		37.79 (9.75) (23–61)						
Sex								
Male	40 (13.3)		67.45 (8.17)	0.767 (0.443)	61.16 (9.01)	−0.062 (0.951)	73.75 (10.18)	1.157 (0.248)
Female	261 (86.7)		66.49 (7.27)		61.23 (6.50)		71.74 (10.18)	
Profession								
Doctor	9 (3.0)		67.03 (11.47)	1.355 (0.250)	61.85 (12.26)	0.820 (0.513)	72.22 (11.66)	1.515 (0.198)
Nurse	225 (74.8)		66.29 (7.09)		61.20 (6.76)		71.39 (9.80)	
Nursing assistant	41 (13.6)		67.60 (6.95)		61.20 (6.76)		73.90 (10.40)	
Medical assistant	20 (6.6)		66.25 (9.56)		59.83 (8.19)		72.66 (12.91)	
Radiation therapist	6 (2.0)		72.77 (4.17)		65.55 (2.72)		80.00 (8.94)	
Marital status								
Single	121 (40.2)		66.63 (7.25)	0.147 (0.863)	61.37 (7.05)	0.560 (0.572)	71.90 (9.96)	0.018 (0.982)
Married	174 (57.8)		66.66 (7.40)		61.22 (6.62)		72.10 (10.35)	
Divorced and widowed	6 (2.0)		65.00 (10.80)		58.33 (10.48)		71.66 (11.49)	
Education level								
High school	25 (8.3)		67.40 (7.51)	0.411 (0.663)	61.73 (5.78)	0.073 (0.929)	73.06 (11.89)	0.710 (0.493)
College	226 (75.1)		66.70 (7.22)		61.17 (6.94)		72.22 (9.91)	
≥Master degree	50 (16.6)		65.86 (8.13)		61.20 (7.15)		70.53 (10.54)	
Regular exercise								
Yes	88 (29.2)		68.18 (6.06)	2.370 (0.018)	62.72 (5.16)	2.452 (0.015)	73.63 (9.70)	1.787 (0.076)
No	213 (70.8)		65.97 (7.80)		60.61 (7.38)		71.34 (10.32)	
Smoking								
Yes	15 (5.0)		67.44 (8.40)	0.188 (0.829)	59.77 (9.12)	0.946 (0.390)	75.11 (10.14)	0.734 (0.481)
No	273 (90.7)		66.53 (7.27)		61.20 (6.65)		71.86 (10.11)	
Ex-smoker	13 (4.3)		67.43 (9.09)		63.33 (8.49)		71.53 (11.98)	
Alcohol								
Yes	141 (46.8)		67.47 (7.24)	1.876 (0.062)	61.67 (6.34)	1.065 (0.288)	73.26 (10.45)	0.299 (0.046)
No	160 (53.2)		65.87 (7.46)		60.83 (7.29)		70.91 (9.84)	
Religion								
Yes	156 (51.8)		65.81 (7.44)	−1.982 (0.048)	60.83 (7.25)	−1.037 (0.300)	70.79 (9.76)	−2.171 (0.031)
No	145 (48.2)		67.49 (7.26)		61.65 (6.42)		73.33 (10.49)	
Living arrangement								
With family	240 (79.7)		66.56 (7.52)	0.278 (0.781)	60.93 (6.88)	1.499 (0.135)	71.13 (9.72)	−0.604 (0.546)
Alone	61 (20.3)		66.85 (6.91)		62.40 (6.72)		72.19 (10.31)	
Economic status								
≥Middle	251 (83.4)		66.19 (7.10)	−2.261 (0.024)	60.87 (6.85)	−2.006 (0.046)	71.51 (9.62)	−1.922 (0.056)
Low	50 (16.6)		638.76 (8.43)		63.00 (6.70)		74.53 (12.43)	
Good sleep quality (PSQI <5)	95(31.6)		68.24 (6.51)	2.612 (0.009)	62.49 (5.85)	2.178 (0.030)	74.00 (9.39)	2.312 (0.021)
Poor sleep quality (PSQI ≥5)	206 (68.4)		65.87 (7.66)		60.64 (7.22)		71.10 (10.42)	

Note. PCS, physical component score; MCS, mental component score; PSQI, Pittsburgh Sleep Quality Index.

**Table 2 medicina-59-00038-t002:** Scores for DASS-21, psychological safety, sleep quality, and health-related quality of life (*N* = 301).

Variables	Min–Max	Mean (SD)
Health-related quality of life	35–90	66.62 (7.39)
PCS	33.33–80	61.22 (6.86)
MCS	36.67–100	72.01 (10.18)
DASS-21 Depression	0–21	4.16 (3.72)
DASS-21 Anxiety	0–21	2.83 (3.22)
DASS-21 Stress	0–21	5.92 (3.72)
Psychological Safety Scale	12–31	22.34 (3.48)
Total PSQI	0–17	6.88 (3.56)

Note. SD, standard deviation; PCS, physical component score; MCS, mental component score; PSQI, Pittsburgh Sleep Quality Index.

**Table 3 medicina-59-00038-t003:** Correlations among DASS-21, psychological safety, sleep quality, and health-related quality of life (*N* = 301).

	Health-Related Quality of Life	DASS-21 Depression	DASS-21 Anxiety	DASS-21 Stress	Psychological Safety	Sleep Quality
Health-related quality of life	−					
DASS-21 Depression	−0.325 (<0.001)	−				
DASS-21 Anxiety	−0.294 (<0.001)	0.781 (<0.001)	−	−		
DASS-21 Stress	−0.354 (0.015)	0.841 (<0.001)	0.762 (<0.001)	−		
Psychological safety	0.100 (0.084)	−0.259 (<0.001)	−0.236 (<0.001)	−0.207 (<0.001)	−	−
Sleep quality	−0.227 (<0.001)	0.388 (<0.001)	0.366 (<0.001)	0.483 (<0.001)	0.006 (0.919)	−

**Table 4 medicina-59-00038-t004:** Results of the stepwise multiple regression analysis for health-related quality of life (*N* = 301).

Variables	Total HRQOL			PCS			MCS		
	B (SE)	*t*	*p*	B (*SE*)	*t*	*p*	B (*SE*)	*t*	*p*
Constant	69.774 (1.821)	38.313	<0.001	60.592 (0.726)	87.786	<0.001	77.017 (2.598)	29.644	<0.001
DASS-21 Depression							−0.950 (0.158)	−6.029	<0.001
DASS-21 Stress	−0.792 (0.106)	−7.457	<0.001	−0.474 (0.104)	−4.548	<0.001			
Economic status	3.929 (1.058)	3.716	<0.001	2.954 (1.041)	−4.548	<0.001	6.045 (1.468)	4.119	<0.001
Alcohol	−1.990 (0.779)	−2.556	0.011				−3.162 (1.069)	−2.959	0.003
PSQI							−0.472 (0.163)	−2.899	(0.004)
R^2^	0.181			0.077			0.203		
Adjusted R^2^	0.173			0.071			0.192		
F	21.943		<0.001	12.489		<0.001	18.859		<0.001

Note. CI, confidence interval, PCS, physical component score; MCS, mental component score; PSQI, Pittsburgh Sleep Quality Index, HRQOL, health-related quality of life.

## Data Availability

The datasets generated and/or analyzed during the current study are not publicly available because they are available from the corresponding author upon reasonable request.

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
