# Peer review of "Factors Affecting Health-Related Quality of Life among Healthcare Workers during COVID-19: A Cross-Sectional Study"

_medicina, 2022, doi:10.3390/medicina59010038_

Round 1

Reviewer 1 Report

The paper is well written, but a few comments need to be addressed. 

Abstract: 

It is essential to include some vital information about your study in the abstract such as the setting and instruments. If your abstract is more extended than it should be, you can delete some information, such as the statical test you used in your study.  

Introduction: 

Your introduction is well written; however, if you include a few studies that examine the same topic before the time of Corna will be better. 

Method: 

No comments 

Results: 

No comments 

Discussion:

 Your discussion is well written; however, if you compare your study with a few studies that examine the time before Corona, it will be better. 

Note: It is acceptable that most previous studies used in the introduction or the discussion were during the time of Coronaviruses; however, I suggest adding a few studies that examine your topic before the virus time, which will make your introduction and discussion more comprehensive. 

Best wishes 

Author Response

Response to Reviewer 1 Comments

Comment #1: Abstract: 

It is essential to include some vital information about your study in the abstract, such as the setting and instruments. If your abstract is more extended than it should be, you can delete some information, such as the statistical tests used in your study.   

Response: We have added and revised the sentences in the abstract.

Background Healthcare workers are threatened by psychological well-being and mental health problems in disasters related to new infectious diseases, such as COVID-19, and they can also have a negative impact on health-related quality of life. Health-related quality of life of healthcare workers should not be neglected because it is closely related to patient safety.

Methods Data were collected from 301 healthcare workers working in five general hospitals with more than 300 beds in two provinces from July 5, 2021, to July 16, 2021. Data were analyzed using SPSS WIN 27.0. The data were analyzed using t-test, one-way analysis of variance, and stepwise multiple regression. Results Our results showed that there was a significant difference in regular exercise, religion, economic status, and sleep quality. The DASS-21 stress, economic status, and alcohol consumption were factors affecting the total health-related quality of life. In the subcategories, the physical component score was influenced by DASS-21 stress and economic status, while the mental component score was influenced by DASS-21 depression, economic status, alcohol consumption, and sleep quality. Conclusions Healthcare workers need continuous and active monitoring of their health level and quality of life, as they are at a risk of increasing work burden and infection due to prolonged exposure to COVID-19, as well as mental health issues such as stress and depression. Additionally, at the individual level, active participation in various programs that can raise awareness of health-related quality of life along with physical health promotion activities should be encouraged. At the organizational level, it is necessary to prepare a compensation system, such as adjusting the workload of healthcare workers and ensuring break time; at the government level, disaster-related policies are needed to ensure a safe working environment for healthcare workers.

Commnet #2 Introduction: Your introduction is well written; however, if you include a few studies that examine the same topic before the time,of it will be better.

Response: Thank you for your comment. Following your advice, we have revised the text as follows.

- In the introduction, we added previous research from page 1, line 38 to page 2, line 53, and described it as follows.

Healthcare workers play an important role in responding to infectious diseases, preventing their spread, and directly engaging in treatment for emerging infectious diseases [2,3,4].

Healthcare workers have different health problems from the general public, and mortality rate from factors other than diseases is high [8]. Moreover, as COVID-19 continues to evolve, treatment protocols have been established, and attempts are being made to address poor psychological well-being, such as mental health problems such as work-related stress and anxiety associated with infectious diseases among healthcare workers [5,9,10,11]. However, emerging infectious diseases, such as Covid-19, further threaten the mental health of healthcare workers and instigate conditions such as depression and post-traumatic stress disorder (PTSD).

- In the introduction, we added previous research from page 2, line 57 to line 60, and described it as follows.

In addition, the consequences of the present pandemic affect not only healthcare workers’ physical health, but also their psychological and mental well-being [1]. Continuous exposure to work-related stress and mental health issues can impair well-being and the ability to work, contributing to poor patient safety, quality of care, and early retirement [5,7]. Furthermore, these problems can adversely affect health workers’ health-related quality of life, including disconnected relationships, problematic use of alcohol and other drugs, and suicidal thoughts. [2,12,13].

- In the introduction, we added previous research from page 2 line 69 to line 82, and described it as follows.

Many healthcare workers experience symptoms of sleep problems, including low sleep quality and short sleep duration, when faced with various threats from emerging infectious diseases [18]. Sleep problems, such as stress and depression, can reduce mental health and work efficiency [19,20]. Insufficient and poor sleep quality can be important early signs of underlying physical or mental health problems for healthcare workers as well as affect their health-related quality of life [20,21,22].

This study aims to identify the relationship between mental health problems, psychological safety, sleep quality, and health-related quality of life of healthcare workers and the factors that influence health-related quality of life during the COVID-19 pandemic.

Comment #3: Discussion: Your discussion is well written; however, if you compare your study with a few studies that examine the time before Corona, it will be better.  

Response 3: Thank you for your comment. Following your advice, we have revised the text as follows.

- In the discussion, we added previous research from lines 288 to 295 on page 10 and described it as follows.

Health-related quality of life differed according to the general characteristics of healthcare workers, regular exercise, and religion. This is similar to the results of previous studies [29,30]. Religion is closely related to psychological aspects and mental health [29], and regular exercise is effective for emotional stability as well as physical health [30]; therefore, it would have been positively correlated with the health-related quality of life of healthcare workers. In particular, regular exercise may help improve fitness, prevent medical conditions, and reduce stress [31]. Therefore, it is necessary to consider activating physical activity and fitness programs for employees.

- In the discussion, we added previous research from lines 345 to 350 on page 12 and described it as follows.

Page12, line 340-345.

On the other hand, reduced sleep quality can impair cognitive function and weaken decision-making ability, reduce clinical work efficiency, and increase the risk of medical errors [21]. Therefore, providing screening and monitoring programs to detect fundamental health conditions and treatments will improve the health of medical workers and improve their health directly or indirectly [22].

Reviewer 2 Report

Dear authors, 

1)"Healthcare workers are at risk of increasing work burden and infection 20 due to the prolonged COVID-19, as well as mental health such as stress and depression." how it is concluded from your study?

2) what about background in the abstract?

3) the aim in the introduction is somehow different from the abstract. 

4) I cannot undrestent the novelty of the manuscript. 

5) the two exclusion crateria is some how repetition of inclusion criteria. 

6) what about sociodemographic data in the method?

7) "The data were collected from healthcare workers working in five general hospitals with more than 300 141 beds in Chungcheng and Jeolla provinces." it is repetetive. 

8) what is the sampling method?

9) "written consent" or "written informed consent"?

10) in first table different types of HCW are not mentioned. 

11) assessing exercise with a yes/no question. How it is possible?

12) the results should be presented according to the aims. 

13) the severity of association should be reported. 

14) "Our results showed that health-related quality of life differed according to general characteristics, including exercise, religion, economic status, and sleep quality. The factors affecting health-related quality of life were stress, economic status, and alcohol consumption." the results in the abstract is confusing. 

15) the discussion and conclusion section should revise accordin to the method section revision. 

Author Response

Response to Reviewer 2 Comments

Comment# 1: "Healthcare workers are at risk of increasing work burden and infection 20 due to the prolonged COVID-19, as well as mental health such as stress and depression." How it is concluded from your study?

Response# 1: Thank you for your comment. Following your advice, we have revised the text as follows.

- In the Abstract, we have revised lines 24 to 32 on page 1.

Conclusions Healthcare workers need continuous and active monitoring of their health level and quality of life, as they are at a risk of increasing work burden and infection due to prolonged exposure to COVID-19, as well as mental health issues such as stress and depression. Additionally, at the individual level, active participation in various programs that can raise awareness of health-related quality of life along with physical health promotion activities should be encouraged. At the organizational level, it is necessary to prepare a compensation system, such as adjusting the workload of healthcare workers and ensuring break time; at the government level, disaster-related policies are needed to ensure a safe working environment for healthcare workers.

Comment# 2: what about background in the abstract?

Response 2: Thank you for your comment. Following your advice, we have revised the text as follows.

- In the Abstract, we have revised lines 9 to 13 on page 1.

Background Healthcare workers are threatened by psychological well-being and mental health problems in disasters related to new infectious diseases, such as COVID-19, and they can also have a negative impact on health-related quality of life. Health-related quality of life of healthcare workers should not be neglected because it is closely related to patient safety.

Comment# 3: The aim of the introduction is somewhat different from that of the abstract.

Response 3: Thank you for your comment. Following your advice, we have revised the text as follows.

- In the Introduction, we have revised lines 80 to 82 on page 2.

This study aims to identify the relationship between mental health problems, psychological safety, sleep quality, and health-related quality of life of healthcare workers and the factors that influence health-related quality of life during the COVID-19 pandemic.

Comment# 4: I cannot undrestent the novelty of the manuscript. 

Response 4: Thank you for your comment. Following your advice, we have revised the text as follows.

- In the introduction, we added previous research from page 1, line 46 to page 2, line 53, and described it as follows.

Healthcare workers have different health problems from the general public, and mortality rate from factors other than diseases is high [8]. Moreover, as COVID-19 continues to evolve, treatment protocols have been established, and attempts are being made to address poor psychological well-being, such as mental health problems such as work-related stress and anxiety associated with infectious diseases among healthcare workers [5,9,10,11]. However, emerging infectious diseases such as Covid-19, further threaten the mental health of healthcare workers and instigate conditions such as depression and post-traumatic stress disorder (PTSD).

- In the introduction, we added previous research from page 2, line 54 to line 60, and described it as follows.

In addition, the consequences of the present pandemic affect not only healthcare workers’ physical health, but also their psychological and mental well-being [1]. Continuous exposure to work-related stress and mental health issues can impair well-being and the ability to work, contributing to poor patient safety, quality of care, and early retirement [5,7]. Furthermore, these problems can adversely affect health workers’ health-related quality of life, including disconnected relationships, problematic use of alcohol and other drugs, and suicidal thoughts. [2,12,13].

- In the introduction, we added previous research from page 2, line 63 to line 67 and described it as follows.

Many healthcare workers experience symptoms of sleep problems, including low sleep quality and short sleep duration, when faced with various threats from emerging infectious diseases [18]. Sleep problems, such as stress and depression, can reduce mental health and work efficiency [19,20]. Insufficient and poor sleep quality can be important early signs of underlying physical or mental health problems for healthcare workers as well as affect their health-related quality of life [20,21,22].

Comment# 5: The two exclusion criteria were repeated inclusion criteria.  

Response: We have deleted the sentence of exclusion criteria.

The exclusion criteria were as follows: 1) Health and medical service personnel who worked in public health centers or industries other than hospitals during the COVID-19 pandemic.

Comment# 6 what about sociodemographic data in the method?

Response: As per the reviewer’s comment, we have added the following sentence.

2.2. General characteristics

The participants’ general characteristics were examined by age, sex, education level, marital status, regular exercise, smoking, alcohol consumption, religion, living arrangement, and economic status. Regular exercise means that adults should exercise for at least 150 min per week.

Comment# 7 "The data were collected from healthcare workers working in five general hospitals with more than 300 141 beds in Chungcheng and Jeolla provinces." it is repetetive. 

Response: As per the reviewer’s comment, we have deleted redundant and repetitive sentences in the data collection.

Page 5, line 160

In the data collection

Data on 301 healthcare workers were collected from July 5, 2021, to July 16, 2021. The survey was completed in approximately 20 minutes and was conducted once. Before proceeding with the study, we explained its purpose to the head of the nursing department and the persons in charge of the departments of the hospitals, and obtained permission to collect data.

Page 5, line 177

In the ethical considerations

This study obtained permission from the directors of nursing and the heads of the nursing departments of five general hospitals.

Commnet #8 what is the sampling method?

Response: As per reviewer’s comment, we revised the sampling method.

Page 2, Line 87

A convenience sample of the respondents of this study were healthcare workers working in five general hospitals with more than 300 beds in Chungcheng and Jeolla provinces.

Comment #9 "written consent" or "written informed consent"?

Response: We have revised the written informed consent.

Comment #10 in first table different types of HCW are not mentioned. 

Response: In table 1, there is no word for HCW.

Comment #11 assessing exercise with a yes/no question. How it is possible?

Response: As per the reviewer’s comment, we have changed the exercise to regular exercise. We have also defined the term.

2.2. General characteristics

The participants’ general characteristics were examined by age, sex, education level, marital status, regular exercise, smoking, alcohol consumption, religion, living arrangement, and economic status. Regular exercise means that adults should exercise for at least 150 min per week.

Comment #12 the results should be presented according to the aims. 

Response: As per the reviewer’s comment, we have revised the text as follows.

4.2. Differences in health-related quality of life

Differences in total health-related quality of life according to the subjects’ general characteristics were analyzed. The total health-related quality of life differed according to regular exercise (t = 2.370, p = 0.018), religion (t = 1.982, p = 0.048), economic status (t = 2.261, p = 0.024), and sleep quality (t = 2.612, p = 0.009). Differences in general characteristics according to the two health-related quality-of-life subdomains were also analyzed. PCS scores according to general characteristics differed depending on regular exercise (t = 2.452, p = 0.015), economic status (t = -2.006, p = 0.046), and sleep quality (t = 2.178, p = 0.030). The MCS, according to general characteristics, differed depending on alcohol consumption (t = 2.299, p = 0.046), religion (t = -2.171, p = 0.031), and sleep quality (t = 2.312, p = 0.021).

4.4. Correlations between DASS-21, psychological safety, sleep quality, and health-related quality of life

Table 3 shows the correlations between DASS-21, psychological safety, sleep quality, and health-related quality of life. Health-related quality of life was negatively correlated with DASS-21 depression (r = -0.325, p <0.001), DASS-21 anxiety (r = -0.294, p <0.001), DASS-21 stress (r = -0.354, p = 0.015), and sleep quality (r = -0.227, p <0.001). Higher health-related quality of life was significantly and weakly linked to lower levels of all the following: DASS-21 depression, anxiety, stress, and sleep quality. DASS-21 depression positively correlated with DASS-21 anxiety (r = 0.781, p <0.001), DASS-21 stress (r = 0.841, p <0.001), and sleep quality (r = 0.388, p <0.001). That is, higher levels of depression were significantly associated with strong positive correlations with higher rates of anxiety and stress, and lower sleep quality. A higher DASS-21 depression score was negatively correlated with psychological safety (r = -0.259, p <0.001). Specifically, the higher the depression score, the lower the psychological safety score. Higher DASS-21 anxiety was associated with higher DASS-21 stress (r = 0.762, p <0.001) and lower sleep quality (r = 0.366, p <0.001), and was associated with lower psychological safety (r = -0.236, p <0.001). Higher DASS-21 stress showed statistically significant negative correlations with less psychological safety (r = -0.207, p <0.001), and higher stress was associated with lower sleep quality (r = 0.483, p <0.001).

Comment #13 the severity of association should be reported. 

Response: We have revised as per reviewer’s comment.

Page 9, line 240

Table 3 shows the correlations between DASS-21, psychological safety, sleep quality, and health-related quality of life. Health-related quality of life was negatively correlated with DASS-21 depression (r = -0.325, p <0.001), DASS-21 anxiety (r = -0.294, p <0.001), DASS-21 stress (r = -0.354, p = 0.015), and sleep quality (r = -0.227, p <0.001). Higher health-related quality of life was significantly and weakly linked to lower levels of all the following: DASS-21 depression, anxiety, stress, and sleep quality. DASS-21 depression positively correlated with DASS-21 anxiety (r = 0.781, p <0.001), DASS-21 stress (r = 0.841, p <0.001), and sleep quality (r = 0.388, p <0.001). That is, higher levels of depression were significantly associated with strong positive correlations with higher rates of anxiety and stress, and lower sleep quality. A higher DASS-21 depression score was negatively correlated with psychological safety (r = -0.259, p <0.001). Specifically, the higher the depression score, the lower the psychological safety score. Higher DASS-21 anxiety was associated with higher DASS-21 stress (r = 0.762, p <0.001) and lower sleep quality (r = 0.366, p <0.001), and was associated with lower psychological safety (r = -0.236, p <0.001). Higher DASS-21 stress showed statistically significant negative correlations with less psychological safety (r = -0.207, p <0.001), and higher stress was associated with lower sleep quality (r = 0.483, p <0.001).

Comment #14 "Our results showed that health-related quality of life differed according to general characteristics including exercise, religion, economic status, and sleep quality. Factors affecting health-related quality of life were stress, economic status, and alcohol consumption." The results in the Abstract are confusing.  

Response: We have revised as per reviewer’s comment.

Page1, line 19

Our results showed that there was a significant difference in regular exercise, religion, economic status, and sleep quality. The DASS-21 stress, economic status, and alcohol consumption were factors affecting the total health-related quality of life. In the subcategories, the physical component score was influenced by DASS-21 stress and economic status, while the mental component score was influenced by DASS-21 depression, economic status, alcohol consumption, and sleep quality.

Comment #15 The Discussion and Conclusion sections should be revised accordingly.  

Response: We have revised as per reviewer’s comment.

In the discussion

Page 11, line 293

Health-related quality of life differed according to the general characteristics of healthcare workers, regular exercise, and religion. This is similar to the results of previous studies [29,30]. Religion is closely related to psychological aspects and mental health [29], and regular exercise is effective for emotional stability as well as physical health [30]; therefore, it would have been positively correlated with the health-related quality of life of healthcare workers. In particular, regular exercise may help improve fitness, prevent medical conditions, and reduce stress [31]. Therefore, it is necessary to consider activating physical activity and fitness programs for employees.

Page 11. Line 340

In this study, sleep quality negatively affected the mental health-related quality of life. This is similar to the results of previous studies [2,7]. Sleep problems negatively affect the immune response by interfering with the daily rhythm of the body, increasing the sensitivity to infection, and having a strong influence on mental health problems such as stress and depression [19,20]. These sleep problems may have reduced the mental health-related quality of life of healthcare workers. On the other hand, reduced sleep quality can impair cognitive function and weaken decision-making ability, reduce clinical work efficiency, and increase the risk of medical errors [21]. Therefore, providing screening and monitoring programs to detect fundamental health conditions and treatments will improve the health of medical workers and improve their health directly or indirectly [22].

Page 12, line 380

In the Conclusion

This study investigated the factors affecting the health-related quality of life of healthcare workers. Physical health-related quality of life was affected by stress, economic status, and mental health-related quality of life was affected by depression, economic status, alcohol consumption, and sleep quality.

Round 2

Reviewer 2 Report

Dear authors, 

the response to the comment 10 "in first table different types of HCW are not mentioned" is not satisfactory. there is no data in the manuscript that who were HCW? how many nurse? how many physicians, and so on.

also the cut point for interpreting of r = 0.38 as strongly, is not acceptable. 

Author Response

COMMNET#1: the response to the comment 10 "in first table different types of HCW are not mentioned" is not satisfactory. there is no data in the manuscript that who were HCW? how many nurse? how many physicians, and so on.

Response: We analyzed and corrected the hcw.

Women comprised 86.7%, nurses comprised 74.8%, married women comprised 57.8% (more than half), and 75.1% were college graduates. 

Commnet#2, also the cut point for interpreting of r = 0.38 as strongly, is not acceptable. 

We deleted the " strong"
